# Some Considerations on the Implications of Protected Areas for Sustainable Development

Alberto Jonay Rodríguez-Darias * and Pablo Díaz-Rodríguez 

Facultad de Ciencias Sociales y de la Comunicación, Instituto Universitario de Investigación Social y Turismo, Universidad de La Laguna, 38200 La Laguna, Tenerife, Spain
* Correspondence: ajroddar@ull.edu.es; Tel.: +034-922317326

**Abstract:** This essay raises some reflections on the implications of protected areas in the processes of social construction related to the conception of nature, its limitations as a strategy for environmental policies (related to ecosystem connectivity and over the management of human activities linked to its functioning), and its public use (encouraging contemplative and tourist uses over productive activities). This essay focuses on some aspects of protected areas as a territorial management model, with the aim of provoking reflection on their implications to sustainable development.

**Keywords:** conservation; nature/culture dichotomy; protected area; public use; sustainable development



## 1. Introduction

A protected area can be defined as a "geographical space, recognised, dedicated and managed, through legal or other effective means, to achieve the long-term conservation of nature with associated ecosystem services and cultural values" [1]. As a territorial management model, it is based on the establishment of boundaries between certain portions of the landscape considered particularly valuable and the rest of the territory, and the development of certain conservation-oriented management strategies inside its limits. This value is generally supported on criteria such as density, rarity, abundance or stability of species, geomorphic structures or ecosystems and scenic beauty, singularity or balance of landscape, as well as levels of concordance with dominant stereotypes about the naturalness, generally according to wild-looking environment.

This being so, it is obvious that protected areas are not 'natural', but the result of a certain model of territorial management. They are an artefact of a relationship with the environment. Considerations about the components of 'the natural' (as opposed to 'the cultural'), their value, the establishment of boundaries and the characteristics of the landscape management model developed within them are all cultural constructs that must be understood as the result of the combination of traditions, conceptualisations, symbolisations, production systems, etc. Both the arguments that give value to the elements considered worthy of conservation and those that justify the location of the boundaries, as well as the determinants of the management model, come from a specific cultural heritage.

The system of protected areas has managed to expand worldwide with notable success. Currently, around 16.64% of terrestrial and inland water, and 7.74% of coastal and marine areas are declared under one of the multiple administrative figures that can be included under the denomination of protected area [2]. At the international level, the percentage of area under environmental protection is expected to grow in the coming decades. In fact, the incremental process of declaring protected areas is often characterised as the path to the adequate conservation of the terrestrial ecosystem, its different manifestations and peculiarities.

However, the success of protected areas as the dominant model for the management of 'the natural' can also be understood as the materialisation of the domination of a particular dualistic worldview of territory ('the cultural' and 'the natural') over other

proposals developed in different places and cultures throughout human history [3,4]. As interventions that enhance specific regulations for the protection of nature and culture [5], certain critical perspectives point to their use as patrimonialisation processes conceived as neoliberal strategies for the profitability of space [6], rather than as tools to promote conservation and quality of life. Classical approaches to explain environmental conflicts in protected areas are mostly based on the differences between native and institutional logics when deciding what is worth to be valued and protected in the territory [7,8]. Faced with them, other perspectives point to the mutual incomprehension derived from the existence of different ontological conceptions that imply the absence of common referents assumed as given [9,10], alluding to the fact that the solution of these problems requires a cognitive justice approach [11].

Although there is consensus in considering that the model of protected areas appeared during the last third of the 19th century in Europe and the United States (with the protection of the Fontainebleau forest, the declaration of Yosemite State Park and Yellowstone National Park), we can find very old examples of spaces in which, for certain reasons (which in many cases could intermingle religious and productive arguments), specific uses and even access were limited. The unequal hierarchy given to these spaces reveals the structural differences in power between symbolic and institutional heritage, evaluated from the prism of a specific political ontology [12] and particular interests beyond conservation, such as tourism or identity [13].

This essay presents some perspectives on the transcendence of protected areas in relation to the symbolic construction of nature, their contribution to conservation and their public use. The aim is not to establish conclusive assertions, but to problematise some implications of protected areas and to encourage reflection on them. We emphasise the contribution of this model to the consolidation and expansion of the nature/culture dichotomy, its limitations as a strategy for environmental conservation (related to ecosystem connectivity and over the management of human activities linked to its functioning), and in the field of public exploitation (encouraging contemplative and tourist uses over productive activities).

## 2. Protected Areas and Cultural Construction of *Nature*

Just as protected areas, nature is a cultural category generated in a specific socio-historical context [14]. Although it draws on earlier sources, the consolidation of the nature/culture dichotomy could be identified in relation to the entrenchment of specific ideals produced in the sphere of European romanticism [15]. Diegues [16] draws particular attention to the influences of these aesthetic ideals on the configuration of the modern stereotype of wild nature, considered extraordinary and especially valuable for having been kept supposedly untouched by humans. It is important to note that the vast majority of these spaces estimated wild from the perspective of the "modern myth of untouched nature" [16] have in fact been the scene of various human activities for millennia [17].

The consolidation of the dichotomous characterisation between nature and culture, as well as the romantic ideals that support it, are related to the profound landscape and cultural transformations that took place in the context of the European industrial revolution, and the growth and population expansion in North America [18]. Among the many contributions that shape this ideal, the influence of poets such as Cleridge, Woodsworth, Carlyle and Ruskin can be highlighted. Along with more systematic works such as *Nature* (published by Ralph Waldo Emerson in 1836 [19]), *Man and Nature: or Physical Geography as Modified by Human Action* (published by George Perkins Marsh in 1864 [20]) or *The Maine Woods* (Henry David Thoreau, 1864 [21]), these authors can be named as decisive examples of the romantic movement that calls for the maintenance of virgin nature as spaces of authentic inspiration; this represents a fundamental seed for the emergence of protected areas as an institutional figure. In fact, the Fontainebleau Forest in France, considered to be the first natural reserve worldwide by special legal act, was declared in 1853 through the proposal of the painters of the Barbizon School [15]. In order to illustrate these ideals, attention

may be drawn to the following excerpt from "The queen of the air" [22], in which we can glimpse the nostalgia for a more authentic past, which has been sullied by the imperatives of progress and production systems.

> *This first day of May, 1869, I am writing where my work was begun thirty-five years ago, within sight of the snows of the higher Alps. In that half of the permitted life of man, I have seen strange evil brought upon every scene that I best loved, or tried to make beloved by others. The light which once flushed those pale summits with its rose at dawn, and purple at sunset, is now umbered and faint; the air which once inlaid the clefts of all their golden crags with azure is now defiled with languid coils of smoke, belched from worse than volcanic fires; their very glacier waves are ebbing, and their snows fading, as if hell had breathed on them; the waters that once sank at their feet into crystalline rest are now dimmed and foul, from deep to deep, and shore to shore.* (John Ruskin, 1869).

In this context, together with the processes of shaping European and American nationalism, the motivation to conserve specific particularly representative elements of the past increases, activating the contemporary process of declaring and conserving cultural heritage [13,23]. As expected, this longed-for past is nothing more than a construction made from the present. Through this process, the past is synthesised, extolling some aspects and obviating others according to the perspective of the person claiming it [24,25]. Certain landscapes thus become key symbolic tools to advocate nationalism [13]. Therefore, "objects, practices or traditions are not a heritage waiting to be discovered, but 'become' through a series of performances" [10,26].

Throughout history, the concept of heritage has undergone an important process of transformation. A dynamic of change that reveals the assumption of some elements over others as worthy of being constituted as relevant elements of nature or culture and shows the nature of heritage as a social construction [27]. This cannot be understood without considering how it is constructed, that is, how the requirements for some elements to become patrimony are agreed and adapted. In this sense, given that heritage is the result of social intervention, it would be better to speak of patrimonialisation [16,28].

Protected areas are a particular form of patrimonialisation [29]. Considered natural heritage, they are indebted to the same sense of longing in the face of landscape transformations driven by the process of industrialisation. Just as cultural heritage aims to preserve and reproduce certain versions of the past, protected areas are oriented towards the performativity of specific ideals about the natural [25]. These symbolic constructions of the value of nature are based both on the coherence of the landscape with particular dominant aesthetic criteria [30] and others based on aspects such as the rarity, abundance, diversity or stability of biological species or geological structures, as well as the representativeness, uniqueness or balance of ecosystems [31,32]. Thus, by making visible its specificity and its link to these categories, its value and need for conservation are stipulated, contributing to reinforcing an image of what should be regarded as true nature.

The recognition of a virginal and immutably balanced nature permeated the first decades of declaration, planning and management of the protected area system [33]. Perceived as eminently primitive, wild and pristine spaces, a model of protected area management for preservation purposes was proposed. It was understood that human actions necessarily constituted a threat to natural processes, which meant that they had to be controlled or eradicated. In fact, the selection and visibility of certain elements of the environment as 'natural' values based on these ideals normalises the criteria for their protection and legitimises the limitation of practices, now understood as conservation measures [34]. From this perspective, it is considered possible to physically delimit natural and cultural spaces, which would be well-differentiated and distinguishable almost at a glance.

Natural space was characterised as valuable in itself, in constant equilibrium only altered by human action [14], in such a way that an important part of the management of 'natural' conservation was based on the containment of specific 'cultural' manifestations. This is consistent with the romantic ideas that inspire the process of generating protected

areas, defending the need to establish natural spaces as sanctuaries of inspiration, as opposed to concerns about the rapid transformation derived from a utilitarian perspective on nature. In this way, protected areas were conceived from the outset as spaces for public use and, specifically, for tourist exploitation [23].

This first phase in the historical development process of protected areas that began with the romantic ideals lasted until the middle of the 20th century. It is a period marked by the consolidation of the first paradigms based on a preservationist perspective. A second phase marked by the expansion of natural reserves at the international level stands out. A fundamental milestone that explains this stage of growth is the foundation of the International Union for Conservation of Nature in 1948. The development of environmentalism and the dissemination of the importance of environmental issues among the different social strata is key to understanding the concern of nations to declare protected areas. In this sense, the influence of some fundamental works such as *A Sand County Almanac* (published by Aldo Leopold in 1949 [35]), *Silent Spring* (Rachel Carson, 1962 [36]) or *The Population Bomb* (published in 1968 by Paul Ehrlich [37]) can be highlighted. Protected areas are commonly perceived as "islands of conservation, of great scenic beauty where urbanites can interact, appreciate and revere wild nature" [4,16]. It is often assumed that protected areas represent what little natural and wildness remains in the world [38].

In this way, protected areas are not only a result of the dichotomous categorisation between 'the natural' and 'the cultural'. The territorial management model based on the declaration of these figures constitutes a fundamental pillar to show this dichotomous perspective, serving as a support for its consolidation and expansion. This is the first of the contributions of protected areas to which we refer in this paper, precisely for their function in materialising, exemplifying and supporting a dichotomous perspective on nature and culture.

Just as other oppositions, the nature/culture dichotomy has been strongly criticised from a poststructuralist perspective at least since the last quarter of the last century [39]. Numerous contributions have made it clear that the boundaries between 'the natural' and 'the cultural' are much more blurred than was considered from the Western perspective [4,9,10,12]. On the one hand, no cultural element is absolutely alien to nature. On the other hand, there are currently not explicitly natural, virgin or wild environments [23], even less so in those declared protected areas, as their very declaration, delimitation and management constitute cultural acts. Currently, "there are no social systems without nature, and few ecosystems without people" [40,41]. Indeed, from the dominant perspective, the technical consideration of wilderness implies the conservation for at least 500 years of more than 70% of its habitat extent and a maximum density of five people per square kilometre (currently represented globally by the Amazon, Congo, New Guinea, the Miombo-Mopane forests, and North American deserts [40]).

Criticism and recognition of the necessity to overcome this dichotomy has been transferred, at least partially, to the field of protected areas, giving rise to new conceptualisations and management strategies [12,41,42]. Thus, a third phase in the historical development process of protected areas could be identified, which expands from the 1990s to the present day. The key feature of the new approaches corresponds to the conviction that protected areas are embedded in wider territories—those ecosystems of which they are part—that need to be included in management [39,43], as well as the importance of reformulating governance strategies aimed at protecting the rights of communities linked to these spaces [44,45].

## 3. Protected Areas and *Nature* Conservation

The declaration of protected areas is the most widespread international strategy to address biodiversity preservation [46]. Classic conservation approaches focus on the preservation of biodiversity, recognition of emblematic rare species and wild-looking landscapes [47]. A protected area harbours important biodiversity niches [48], which in many cases, involve the protection of environments related to environmental services,

such as oxygen production, carbon dioxide metabolism, soil protection, mitigation of desertification processes, water generation, adaptation to the impacts of climate change, buffering against the effects of natural disasters, etc. [49,50], and their potential to confront problems derived from socioecological processes such as globalisation, migrations or individualisation [41,51].

Global comparative studies show encouraging results that emphasize the importance of these figures of protection. The research of Cazalis et al. [52] along eight tropical forest biodiversity hotspots around the world (Atlantic Forest, Tropical Andes, Tumbes-Chocó-Magdalena, Mesoamerica, Eastern Afromontane, Western Ghats and Sri Lanka, Indo-Burma and Sundalad) and in North American forests [53] provides evidence of the benefits driven by protected areas in preventing both forest loss and degradation and protecting the diversity of bird species in some of the world's most diverse and threatened terrestrial ecosystems. In the same sense, global meta-analyses on a local scale show the positive effects on biodiversity worldwide, reinforcing the global importance of these figures of protection [54,55]. Moreover, among other success cases the investigations carried out by López-Angarita et al. [56] show us the usefulness of protected areas in stopping the deforestation of mangroves of the eastern tropical Pacific, enhancing sustainability and contributing environmental services to the development of the region. Riggio et al. [57] highlight the usefulness of protected areas to conserve habitats in East Africa. Similar examples can be found in other rich ecosystems, such as Antarctica, where analyses support the effectiveness of protected areas to improve biodiversity protection across the continent [58].

Protected areas have been characterised as sanctuaries for the protection of ecosystem values. They constitute refuges and networks [59] for some species or habitats that are particularly sensitive to the effects of certain human activities or climate change [60], and so contribute to preventing the extinction of species [61]. Their contribution is also highlighted insofar as they help to control the overexploitation of species (both terrestrial and marine, animal and plant), thus becoming safe spaces for their reproduction [62]. Although, in this regard, it is important to pay attention to the reproduction or regeneration targets for habitats, as Plumeridge and Roberts [63] show for the marine realm, sometimes these targets are unambitious, being able to consider as regeneration success the recovery of a stock of marine resources much lower than the effectively potential one.

The territorial management model based on protected areas implies a public intervention in practice comprising two basic mechanisms: the establishment of their boundaries and the development of a management model fundamentally oriented towards the conservation of their natural values.

The process of delimiting protected areas is always complex and can be based on a multitude of possible criteria. Issues such as the visual continuity of the landscape, the existence of natural or artificial boundaries (ravines, rivers, roads, buildings, etc.), the main location of concrete species deemed to be of special value, ecological connectivity, political interests or urban development, among many other possible criteria, can be taken as unique criteria or, more commonly, intermingled in the configuration of protected area boundaries. This is not a trivial matter; this process marks the frontier between 'the natural' and 'the cultural'. It marks the limit between the space that should be especially oriented to the conservation of its natural values (worthy of environmental conservation) and that which is more dedicated to the development of different human activities.

The establishment of these boundaries implies problematic issues for general conservation objectives. Given the systemic nature of ecological processes, it does not seem possible to confine them to borders that have more to do with administrative and legal issues than with ecosystem functioning. Although, as mentioned above, protected areas can play an important role as a refuge for individuals of certain species, they do not, in general, constitute measures that are coherent with the phenomena and processes of interdependence and connectivity (both vertical and horizontal [64]) of protected spaces with others outside their boundaries [43,65,66]. Furthermore, the static nature of the location of

protected areas does not correspond to the dynamism of ecosystem networks [58] or the mobility of a significant number of species under protection [67].

The recognition of this problem has led to a profound debate, leading to management proposals with emphasis on the porous character of the boundaries of protected areas and the need to manage them in relation to their environment instead of continuing to reproduce the nature/culture dichotomy [68,69].

The territorial management model based on protected areas implies the development of plans and strategies aimed at the conservation of natural values within them. Although they have shown great capacity as a key tool for the conservation of nature and, with it, biodiversity, as a reference to mitigate climate change, they are not exempt from complications. Firstly, it should be borne in mind that there are numerous cases in which, following the declaration of a protected area, the instruments, funding or material and human resources for its proper management are not developed [61]. Conservation tasks imply relatively high costs that, on many occasions and especially in developing nations, are not met, giving rise to so-called 'paper parks' [70].

Beyond the problems related to the effective implementation of the measures, certain difficulties related to the conception of the conservation strategies themselves can be highlighted. It has already been mentioned that the traditional conceptualisation of protected areas is strongly related to the classic dichotomy between 'the natural' and 'the cultural', considering that these spaces fall under the realm of 'the natural'. However, the reality is undoubtedly more complex than this Cartesian characterisation of reality suggests. The vast majority of landscapes in protected areas are the result of the coexistence of human activities with other ecosystemic processes and relationships [17]. Forestry, livestock, logging, wild fruit gathering, hunting, fishing, shellfishing, etc., have been carried out in many of these areas for centuries, on many occasions, under conditions that have allowed the conservation and reproduction of the ecosystemic values that recently led to their declaration as a protected area. In this way, it can be understood that these human activities have been integrated into the ecosystem processes as one more variable among those that explain the presence of a given landscape and environmental conditions [71,72]. From a classical perspective, the management of protected areas has tended to displace these human activities [73,74] because, in line with the nature/culture dichotomy, it is assumed that they constitute a threat 'to nature' [75]. There are clear exceptions to these processes; especially in the field of marine protected areas, the benefits that fishing communities can obtain as a consequence of controlling overfishing, the increase in fishing stocks, the limitation of certain fishing gears and the improvement of environmental conditions are noteworthy [76,77]. The local development opportunities derived from a greater stock of natural resources or those derived from some tourism models are also noteworthy [78].

The implications of these considerations for the development possibilities of communities traditionally linked to spaces declared protected areas have been discussed at length [14,79]. Three decades ago, in the IV World Parks Congress, in the "Peoples and Parks" working document [80] or the "Action Plan for Protected Areas in Europe" [81], there was already recognition that the environments declared protected areas are not strictly natural, but that throughout history they have suffered (or enjoyed) a greater or lesser degree of contact with human populations. In these same spaces of debate and documents, there is also an acknowledgement of the problems generated by the dominant paradigm around protected areas for the socio-economic development of populations whose way of life depended on the productive use of protected areas [41,82]. Undoubtedly, the appropriate conservation of 'wild nature' is an ethical principle and can provide fruitful socio-ecological benefits. However, there are sometimes significant tensions in the relationship between this objective and socio-economic development, as well as with the symbolic link of local populations with these spaces, leading to the creation of zoos or theme parks for the sake of 'wilderness' and 'naturalness'.

As mentioned above, these issues have been highlighted for at least three decades. They have led to interesting proposals aimed at integrating conservation and local develop-

ment objectives with governance-based perspectives [83–86]. Even with the recognition of these problems, protected area management policies are still often based on traditional perspectives and reproduce situations of discrimination against local communities [87]. Their impact on socio-environmental networks and the life chances of certain populations are incontestable; it was estimated that this model has generated 130 million refugees [74]. Critical approaches call for governance structures and participation of local populations, warning of the dangers of the perverse use of these figures as a re-territorialisation mechanism for the sake of the capitalisation of spaces, according to some neoliberal strategies linked to particular conceptions of nature conservation [88,89]. However, there is also evidence that protected areas can improve the social conditions of local populations facing poverty [90].

Numerous examples have shown the importance of the participation and co-management of populations linked to protected areas for the achievement of greater socio-ecological success. Dawson et al. [91] reviewed 169 cases examining the role of indigenous people and local communities' influence on conservation outcomes in protected areas around the world. They found that most positive effects for quality of life, biodiversity conservation or ecosystem restoration come from contexts where indigenous people and local communities play a central role in decision making and local institutions are a recognised part of governance. Trujillo-Arias et al. [92], for example, through a multi-temporal analysis of the forest dynamics, highlight the importance of a peasant reserve zone in the Magdalena Valley region (Colombia) as a successful example of community management in the conservation of a tropical humid forest and endangered species.

The potential of partnerships between indigenous communities and government institutions for the conservation of global diversity has been proven from the analysis of indigenous-managed lands and existing protected areas in Australia, Brazil and Canada [93]. The results show similar levels of vertebrate biodiversity between both areas, which could be improved through a mixed use of both types of management that recognize indigenous leadership in conservation processes. Likewise, Coggins et al. [94] reveal the importance of the relationship between the conservation status of China's village fengshui forests and their spiritual significance. Considered by local populations as symbols of good fortune, these forests are customarily conserved without official recognition in most cases despite their socioecological importance, their function as a biodiversity refuge, and their key role in large-scale conservation networks.

Other figures worth mentioning in this sense are the so-called exclosures. This practice is a territorial intervention consisting of excluding certain areas from the land use in order to promote natural regeneration and reduce degradation in communal grazing lands. The case of Ethiopia has been widely studied, where its usefulness for preserving biodiversity in highly degraded lands [95] or restore environmental conditions of the lands has been demonstrated [96], specifically improving the woody species diversity and soil chemical properties [97]. Therefore, it can be a strategic instrument to influence carbon storage and enhance livelihood conditions in the long term, especially when local knowledge and governance are considered in the decision making [98].

It is essential to effectively include inhabitants and productive activities in the decision making, design and management models [99,100], as well as give relevance to symbolic representations of the territory as a tool for shaping the landscape [101–103]. For this, it is necessary to address the implications derived from the power asymmetries inherent in conservation and co-management of protected areas when different relational ontologies are involved. This requires an epistemological effort from the perspective of cognitive justice that recognises ontological diversity and overcomes the simple assumption of different cultural identities [11].

## 4. Public Use of Protected Areas

An important contribution of the international system of protected areas is its potential for positively affecting well-being and healthy ageing [104–106] as well as the development

of recreational and environmental education activities [107]. In fact, it is no coincidence that the most widespread name for these types of space is "parks". This seems to fit in with the conception of parks as spaces for recreation and admiration, in the manner of extensive gardens. Since the beginning of this territorial management model, the objective of public use has been, together with that of conservation, one of the most recurrent goals in the documents for the declaration and management of protected areas. As early as the declaration of Yellowstone National Park, the United States Congress designed it as a "public park or recreation area for the benefit and enjoyment of the people" [108].

The opportunities for education and other benefits for the quality of life of populations linked to the consumption of the natural landscape and the development of activities in this environment are evident [109]. Benefits include sport and health promotion [110], in situ educational situations, improved environmental sensitivity, aesthetic enjoyment, etc. The potential of the development of activities in this environment for the improvement of cognitive, affective and social performance in children has even been demonstrated [111].

Protected areas constitute spaces in which one can feel and experience a special relationship with environmental values. This, in addition to facilitating the development of activities that result in improving the quality of life of the populations, offers privileged situations for the development of awareness-raising campaigns. Through different means, such as visitor centres, protected areas serve as a basis for increasing public awareness of environmental values, threats and the importance of conservation [112].

Interest in visiting these sites has grown significantly in recent decades, especially in recent years, following the periods of confinement and the different situations caused by COVID-19 [113,114]. This represents both an opportunity and a challenge in the planning and management of protected areas and the conservation of their fundamental ecosystem values.

Among the different options for public use in protected areas, their strong link with tourism stands out. The popularisation of the romantic use of forests through outdoor recreation and tourism beginning in the 19th century was a key trigger for the invention of nature conservation through modern protected areas [115]. From the growing appreciation of the sublimity of nature in contrast to the sphere of culture as Western ontological categories, tourism began to be configured as part of conservation strategies [115].

In an increasingly urban world, environments less modified by human action play an increasingly important role in providing aesthetic pleasure and recreation to the population [116]. As expected, this need for the natural (the summa of which is identified with those spaces with lush vegetation, high density of fauna—especially if they are species with positive cultural value—and biodiversity) has been detected and used (or directly created) for commercial purposes; among them, for tourism [117,118]. In fact, as shown in Figure 1, the process of declaring protected areas and tourism at the international level are two phenomena that have historically evolved in parallel. The similarity between both trends is evident and, although in principle they could be understood as independent phenomena, there are reasons to defend the relationship between both variables. Following the argument of Simancas [119], in the case of the protected areas of the Canary Islands (Spain), the relationship between the motivations for declaration and tourism occurs in two senses: 'due to tourism' (as a strategy to defend specific environmental values in the face of tourism development) and 'for tourism' (linking the declaration with the increase in tourist interest in the destination [120], including the incorporation of certain changes in its projected image and its inclusion in particular international promotion networks).

In recent decades, the evolution of international tourism trends has favoured an increase in the demand for nature tourism products, including those related to protected areas. On the one hand, the increase in the percentage of the urban population seems to imply dissatisfactions (related to lifestyle, landscape, air quality, etc.) which generate the need to move to areas where the transformations generated by human action are less visible or where they conform to ideals related to 'nature'. On the other hand, for certain sectors of the population, the traditional forms of mass tourism have lost their appeal in

the face of the new forms of tourism that have been consolidated since the second half of the 1990s [121].

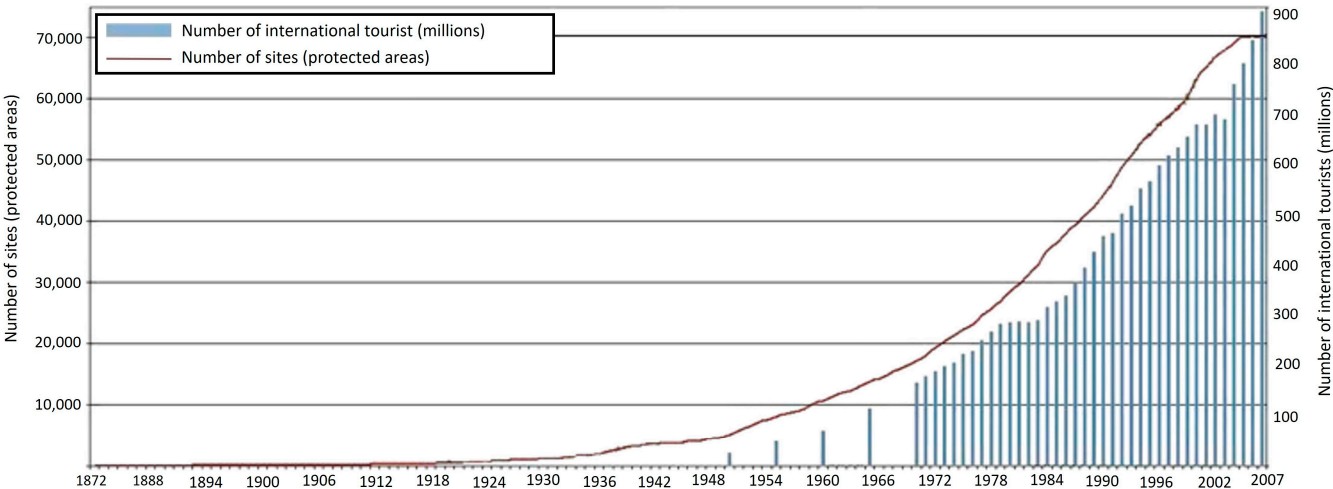

**Figure 1.** Evolution of the number of tourists and protected areas at the international level between 1872 and 2007. Source: own elaboration based on data from the Global Policy Forum, UNEP and the World Tourism Organisation.

The tourist use of protected areas has often been identified with ecotourism. From the initial definition of Ceballos-Lascuráin [122], this is defined as the visit to areas regarded natural with the motivation of enjoying the environment. Much has been written in recent years about the tourist uses of protected areas, although, in most cases, these approaches have been based on the consideration that tourism in these environments responds to sustainability criteria [123]. From a critical point of view, it can be considered too limited and may even have been influenced by the development of specific products associated with marketing trends aimed at attracting the attention of concrete tourist profiles. Within the scope of this work, perspectives based on tourism consumption of protected areas in a more general context, without assuming per se their accommodation to sustainability criteria, are conceived as more coherent with reality. The fact that these are visits and activities carried out within protected areas does not guarantee their adequate management or the absence of highly negative impacts. In fact, tourism in protected areas manifests massive influxes in certain environments, and the impact of these can be much more detrimental than in other areas specially designed for conventional tourist use [124]. Marketing strategies aimed at segmenting demand have made it easier for a tourist visiting a protected area to experience their trip as more sustainable than that of a traditional sun and beach tourist, but the degree of sustainability of one or the other option will depend on a multitude of variables, not on the type of attraction that motivates tourist consumption.

The implications of tourism commercialisation of protected areas can be very diverse and complex. In relation to economic aspects, it is common to highlight the effects that tourism can have on local populations [125] and the financing of conservation strategies [117,126]. On the one hand, the incorporation of tourism is seen as an alternative source of employment and income for the populations that traditionally use the area. This can compensate for the restrictions on the productive use of the area derived from conservation-oriented management strategies. At the same time, it can be a stimulus for conservation, either from a symbolic point of view, which is based on a supposed increase in the valuation of the natural heritage by the resident population as they understand it to be valued by others (tourists), or from a more practical perspective, based on the perception of the interdependence between the conservation status of areas and the economic benefits resulting from their exploitation as a tourist resource [44,127].

It is necessary to highlight that the tourist use of protected areas can have important impacts on specific species or ecosystem processes. These can be derived from the wide

range of direct impacts that the visit can generate (introduction or extraction of biological material, soil erosion, generation of waste, pollution, etc.) or from the adaptation of the environment to attract them (which includes both the construction of infrastructures and the artificial reproduction of specific landscape aspects with great tourist attraction [128]).

In the socio-cultural sphere, the implications of the commercialisation of protected areas for tourism can also be very varied. On the one hand, tourism is a catalyst for social change processes, especially in terms of social norms, perception of others, moral conduct, health, religion, status of individuals, clothing, food, leisure activities, etc. [118,129]. On the other hand, the process of commercialisation of protected areas frequently involves strategies of resignification of the territory to adapt it to the standards of visitors, which can translate into conflicts and undesirable effects for other population groups, as well as their identity, self-esteem and forms of interaction with the territory.

In this sense, tourism development can lead to the abandonment of other productive activities and produce dependence, which implies uncertainty on a volatile sector dependent on global strategies that transcend the possibilities of local control. In this sense, different case studies show the suitability of betting on tourism as a strategy to complement traditional productive activities, avoiding its consideration as an alternative to them [130].

On the other hand, the processes of naturalisation of protected areas to adapt their characteristics and meanings to the romantic imaginary of tourists imply a discursive re-symbolisation of the environment and the populations that inhabit it. These processes linked to the creation and management of protected areas have important repercussions in relation to access to resources and the power structures inherent to these contexts, and can act as tools to legitimise or delegitimise the role played by the social actors who inhabit protected areas [7,8]. As Sobhani et al. [131] show in a recent case study in Iran, local communities can be a strategic stakeholder. The results of this analysis suggest that the degree of sustainability of tourism in protected areas correlates positively with the levels of participation of these communities.

Thus, although protected areas have the potential to promote sustainable development and improve the quality of life of populations through different forms of public use, the success of these objectives will depend on the effectiveness of planning and management strategies [132,133].

## 5. Final Considerations

This essay invites us to reflect on the social construction that implies the assumption of the constituent elements of a protected area, which condition the establishment of its limits and the assumption of certain uses of the territory based on their perception.

The model of land and resource management based on the declaration and management of protected areas has shown great potential to address the challenge of conserving species and ecosystem processes, as well as the challenge of climate change. The socio-cultural benefits of protected areas can also be wide-ranging. In addition to their use as spaces for leisure and recreation in contact with 'nature', the conservation of the territory through protected areas promotes the reproduction of a multitude of environmental services that favour sustainable development; they constitute scenarios in which to develop traditional productive activities with high ecological value and can facilitate economic alternatives linked to ecotourism in the context of rural populations traditionally excluded from the production possibilities offered by urban and industrial environments.

As has been indicated, the concept of a protected area is related to a peculiar territorial perspective, identifiable with the biocratic style of conservation [134], which maintains a dichotomous relationship between natural and anthropized spaces. This ignores the fact that an important part of the spaces considered natural are the result of the centuries-old interaction of environmental characteristics with human practices linked to agriculture, livestock and forestry, among others. To the establishment of these frontiers, this perspective adds the conception that natural spaces must be subject to specific management that guarantees the conservation of their ecological values, while anthropized environments

can continue to reproduce the developmentalist model that characterises the evolution of the capitalist production system.

It is in relation to the latter that we can once again raise a critical perspective. Protected areas favour the sustainability of development, insofar as they allow for its self-reproduction and have served as a strategy to lessen attention on particularly worrying effects, as well as to attenuate the demands of social movements that are raising their voices in defence of more sustainable forms of development. From this point of view, the process of declaring protected areas should not be understood as a strategy opposed to capitalist models of production and consumption. Rather, this model could be defined as a corrective measure for some self-destructive effects detected in the forms of production and consumption that have spread after Fordist development.

From a critical perspective, protected areas may play a significant role in the processes of legitimisation, diversion of attention and reproduction of certain unsustainable models of production and consumption. Protected areas may even enhance particular unsustainable forms of nature-based tourism. However, the real threats to global sustainability transcend the limits imposed on these spaces, which can certainly be characterised as islands of conservation in vast oceans of environmental destruction.

**Author Contributions:** A.J.R.-D. and P.D.-R.: conceptualisation; writing—original draft preparation, review and editing; visualisation; supervision; funding acquisition. All authors have read and agreed to the published version of the manuscript.

**Funding:** This research was funded by the project "El equilibrio territorial post-COVID-19 en Canarias. Nuevas estrategias para nuevos tiempos. PROID2021010026", through the Research and Innovation Strategies for Smart Specialisation (RIS3) of the Canary Islands.

**Institutional Review Board Statement:** Not applicable.

**Informed Consent Statement:** Not applicable.

**Data Availability Statement:** Not applicable.

**Conflicts of Interest:** The authors declare no conflict of interest.

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
