# Peer review of "Some Considerations on the Implications of Protected Areas for Sustainable Development"

_sustainability, doi:10.3390/su15032767_

Round 1
Reviewer 1 Report (Previous Reviewer 4)
Revisions were helpful, especially the added citations and examples.
Author Response
Thank you very much for your comments. The manuscript has improved thanks to your help.

Reviewer 2 Report (New Reviewer)
This essay is relatively well written. But the authors pay attention mainly to the public side of implications of protected areas. In contrary, I believe that the authors need to demonstrate also scientific side of public implications of the protected area idea. It is well known that many organisms are preserved within protected areas in various regions of the world. I strongly recommend to highlight the importance of protected areas in the global scale with cases from national cases.
In addition, please, consider that there are some allies of protected areas in various parts of the world. For examples, Trujillo-Arias et al. (2023: https://ncr-journal.bear-land.org/article/397https://dx.doi.org/10.24189/ncr.2023.003) highlight the importance of a campesine reserve zone in the Magdalena Valley (Colombia) in the conservation of threatened species. Another case is so-called exclosures, when some areas are being excluded from the land use to preserve biodiversity (https://doi.org/10.1016/j.heliyon.2021.e06898) or restore environmental conditions of the lands (http://dx.doi.org/10.24189/ncr.2018.001). I think the authors can find other cases.
After these corrections, the role of protected areas will be highlighted in the more complete level.
Author Response
Point 1: This essay is relatively well written. But the authors pay attention mainly to the public side of implications of protected areas. In contrary, I believe that the authors need to demonstrate also scientific side of public implications of the protected area idea. It is well known that many organisms are preserved within protected areas in various regions of the world. I strongly recommend to highlight the importance of protected areas in the global scale with cases from national cases.
Response 1: Thank you for your suggestion.
It is true that we have emphasized the social construction of the values that underlie protected areas. We have compensated for this perspective with examples in which, indeed, these figures have contributed positively to conservation and preservation.
We have added important study cases and global comparative studies, in addition to those that were already, reinforcing the global importance of protected areas (lines 227-235; 240-243).
We have also included other examples of success cases that emphasize the role of local populations in achieving socio-environmental benefits through protected areas (see point 2).
Point 2: In addition, please, consider that there are some allies of protected areas in various parts of the world. For examples, Trujillo-Arias et al. (2023: https://ncr-journal.bear-land.org/article/397https://dx.doi.org/10.24189/ncr.2023.003) highlight the importance of a campesine reserve zone in the Magdalena Valley (Colombia) in the conservation of threatened species. Another case is so-called exclosures, when some areas are being excluded from the land use to preserve biodiversity (https://doi.org/10.1016/j.heliyon.2021.e06898) or restore environmental conditions of the lands (http://dx.doi.org/10.24189/ncr.2018.001). I think the authors can find other cases. After these corrections, the role of protected areas will be highlighted in the more complete level.
Response 2: Thank you for your comment.
We believe that the aspect you mention was not sufficiently explained in the manuscript.
Following your recommendations we have included several key global and local examples showing the importance of populations in protected area management, as well as other models that can be considered allies of protected areas (lines 346-374; 511-514).
Thanks to your comments the manuscript has been substantially improved and now presents a broader and better explained casuistry in relation to the role of protected areas.
Reviewer 3 Report (New Reviewer)
The manuscript is well structured, and the conclusions are based on sufficient analysis of results. Generally, I found this study is relatively novel and fits well into the scope of Sustainability. However, I found some very important issues and biases which need to be addressed before considering for acceptance.
Major comments
- Some sentences are not clear to me; the authors should explain better the meaning of the following sentences:
L 20-23
L 48-50
L 134-135
L 179-180
- The literature review is not comprehensive. Some recent citations about the topic of "protected areas for sustainable development" has not been added (I am not one of the authors of the following citations), e.g.:
** https://doi.org/10.3390/land11101871
** https://doi.org/10.3390/su141710956
** https://doi.org/10.3390/f13050740
** https://doi.org/10.3390/w14244121
- The English writing is readable but still has room to be improved. The improvement in writing might not take much time for the authors, but it can significantly improve the presentation quality and increase its impact. My main concerns are related to improper paragraph structure, grammatical inconsistencies (e.g., use of was, is and has been in the Introduction) and repetitions throughout the manuscript.
Minor comments:
L 122- 126: Remove this sentence. This is unnecessary and redundant.
L 130: Authors briefly discuss the patrimonialisation.
L 173: I still do not understand this model: "territorial management model"
L 346-348: Is it true for national visitors? or for national and international visitors?
Author Response
Point 1: The manuscript is well structured, and the conclusions are based on sufficient analysis of results. Generally, I found this study is relatively novel and fits well into the scope of Sustainability. However, I found some very important issues and biases which need to be addressed before considering for acceptance.
Response 1: Thanks for your comments. They have helped us to rethink the manuscript. We believe it has been vastly improved in line with your expectations.
Point 2: Some sentences are not clear to me; the authors should explain better the meaning of the following sentences: L 20-23.
Response 2: Thank you. We have revised the sentence and we believe that it can now be better understood (lines 21-27).
Point 3: L 48-50
Response 3: Thank you. We have changed the sentence and we think it is clearer now (lines 55-60).
Point 4: L 134-135
Response 4: Thank you. We have changed the sentence and now it can be better understood (lines 144-148).
Point 5: L 179-180.
Response 5: Thank you again for your comments. It has made us realize that referring to these examples of classical dichotomies would open a debate that exceeds the scope of this manuscript and could lead to losing sight of its relevant aspects. So we have decided to remove them from the text.
In any case, since the consolidation of the poststructuralist paradigm, culturally established dichotomous classifications have been strongly questioned, showing that they are social constructions. Precisely the nature/culture opposition is one of them, as can be seen in the debate on the ontological turn that continues today.
Point 6: The literature review is not comprehensive. Some recent citations about the topic of "protected areas for sustainable development" has not been added (I am not one of the authors of the following citations), e.g.:
** https://doi.org/10.3390/land11101871
** https://doi.org/10.3390/su141710956
** https://doi.org/10.3390/f13050740
** https://doi.org/10.3390/w14244121.
Response 6: Thank you very much for the proposal and for the references you have provided.
We have expanded the literature review to include important case studies, including the recommended authors. In addition, global comparative studies have been added (lines: 227-236; 240-243; 346-374; 511-514).
Although the aim of this manuscript is not an exhaustive analysis of the literature, but a dissertation on certain points of view based on theoretical references and case studies, we believe that it now presents a broader and better explained casuistry in relation to the role of protected areas.
Point 7: The English writing is readable but still has room to be improved. The improvement in writing might not take much time for the authors, but it can significantly improve the presentation quality and increase its impact. My main concerns are related to improper paragraph structure, grammatical inconsistencies (e.g., use of was, is and has been in the Introduction) and repetitions throughout the manuscript.
Response 7: The manuscript has been proofread by a native English speaker. We hope that the English writing has improved sufficiently.
Point 8: L 122- 126: Remove this sentence. This is unnecessary and redundant.
Response 8: Thank you very much, we have deleted the sentence.
Point 9: L 130: Authors briefly discuss the patrimonialisation.
Response 9: Thank you for your suggestion. We have been considering it carefully and we believe that what you are requesting is already stated in the previous paragraph. In accordance with your previous comment (Point 8), we would like to avoid redundancies.
Point 10: L 173: I still do not understand this model: "territorial management model"
Response 10: Thank you for your comment. From our perspective, the establishment of certain limits in the territory based on what is considered particularly valuable from an environmental perspective and to establish a strategy aimed at its conservation (limiting land uses inside the limits and providing a more permissive model abroad) constitutes a certain model of territorial management. This approach is a far cry from other proposals based on a more holistic perspective of the territory.
Point 11: L 346-348: Is it true for national visitors? or for national and international visitors?
Response 11: This is an interesting assessment. Actually the casuistry is very diverse. Globally this assertion is true for national and international visitors. But this varies from region to region. With this sentence we wanted to underline that the demand for visiting natural areas has grown in the last decades, and especially after COVID-19. This is reflected in the growth of nature-based tourism but also in the overall number of visitors to nature sites, regardless of whether the visit involves crossing national borders or not.
Thank you very much for your review. We believe that thanks to your comments the manuscript has been substantially improved.
Reviewer 4 Report (New Reviewer)
This is a sophisticated review of these issues. You might want to more clear on what you are offering with the paper. None of the ideas are new, what new contributions do you wish this manuscript to offer?
Author Response
The manuscript presents a broad, integrative and critical reflection on the main implications of protected areas for development. It assumes a point of view based on certain perspectives that we believe are generally not sufficiently considered as a whole. In particular, we believe that the implications of the processes of resignification and the ontological turn debate when reflecting on the role of protected areas in sustainable development are not widely known or taken into account by certain disciplines. Our vision seeks to problematize certain implications of protected areas and encourage reflection on their role in the consolidation of the nature-culture dichotomy and the limitations that this entails for their environmental conservation objectives and in the field of public use.
We have made a revision of the manuscript presenting a broad and better explained casuistry in relation to the role of protected areas. This essay can also be an introductory reflection for the special issue 'Protected Areas and Their Contribution to Sustainable Development' for the journal Sustainabiliy.
Round 2
Reviewer 3 Report (New Reviewer)
I am happy with the revised version.
Reviewer 4 Report (New Reviewer)
Thank you for making the improvements
This manuscript is a resubmission of an earlier submission. The following is a list of the peer review reports and author responses from that submission.
Round 1
Reviewer 1 Report
· The aim of this manuscript is really unclear and confusing. What do authors imply by Lights and Shadows of the natural management model based on protected areas? and how are those main contributions to sustainable development defined?
· In general, this manuscript is just a mere dissertation based on the opinion of the authors and it is not a further review of updated scientific literature. Authors used very old quotations.
· The abstract says basically nothing of the content of the manuscript. It says that “this paper provides some reflections of the main contributions of the protected areas system” My question here is where? Authors stated that “the paper considers protected areas as a human cultural production…” and then authors state that “The aim of this paper is to reflect on the lights and shadows of the nature management model based on protected areas…” In reality, the abstract does not provide any key findings of the review or any clear ideas of the reflection.
· The Introduction sections addressed none updated literature review. I expected an introduction addressing the scientific mainstream of natural protected areas worldwide, but in reality, it is just a perspective of the authors based on very old literature.
· The section Protected areas and cultural construction of Nature is a mere perspective, without a further approach of scientific and updated evidences of the usefulness of the protected areas worldwide.
· If it is an article reflection about the protected areas, I miss quotations by Leopold and Rachel Carson, for instance, who were promoters of natural protected areas in the US.
· The assertion in section 3, A protected area is considered the most successful instrument for nature conservation and the quotation of the Convention of Biological Diversity is disturbing. Actually, a protected area is not the most successful instrument but one of many other management instruments. Also, the authors fail in quoting the works by Callum Roberts, Tundy Agardy, and many others.
· In sum, the work is a mixture of authors points of view and not a throughout review of updated and scientific evidences either in support or against what a natural protected area, either in terrestrial or marine environments.
· I regret the work is not accomplishing to what is intended to, so I recommend rejection.
Author Response
Point 1: The aim of this manuscript is really unclear and confusing. What do authors imply by Lights and Shadows of the natural management model based on protected areas? and how are those main contributions to sustainable development defined?
Response 1: Thank you very much for your review. We have followed your suggestions and incorporated them into the revised manuscript.
It is true that it is not a systematic analysis of the lights and shadows of protected areas as a nature management model. We believe that one of the main problems with the manuscript has been that it was not sufficiently clear that it is an essay, not an experimental analysis paper. This error has undoubtedly influenced the expectations of its reading.
We think that in the new version It is clarified that it is an essay, and the aim of the paper is now clearer. Clarifications have been included in the abstract, introduction and final considerations (especially in lines 8-13; 59-71; 250-253; 453-455). Furthermore, we have decided to replace the term 'contributions' with 'implications' in the title, since we believe that the former could cause confusion.
As expressed now, the aim is not to establish conclusive assertions, but to problematize some implications of protected areas and to incite to the reflection particularly on the contribution of the protected areas model to the consolidation and expansion of the nature-culture dichotomy, its limitations as a strategy for environmental conservation (related to ecosystem connectivity and the management of human activities), and in the field of public exploitation (encouraging contemplative and tourist uses over productive activities).
Point 2: In general, this manuscript is just a mere dissertation based on the opinion of the authors and it is not a further review of updated scientific literature. Authors used very old quotations.
Response 2: We thank you again for this appreciation, which we believe is related to the previous point.
Indeed, as has been mentioned, it is a dissertation. We consider that in the previous version of the work it was not sufficiently clear that it was an essay.
Without intending to make a state of the art, we have improved bibliographical references and updated our arguments based on more updated scientific literature throughout the manuscript. Thanks to your comments, the number of bibliographical references has increased significantly.
Point 3: The abstract says basically nothing of the content of the manuscript. It says that “this paper provides some reflections of the main contributions of the protected areas system” My question here is where? Authors stated that “the paper considers protected areas as a human cultural production…” and then authors state that “The aim of this paper is to reflect on the lights and shadows of the nature management model based on protected areas…” In reality, the abstract does not provide any key findings of the review or any clear ideas of the reflection.
Response 3: Thank you for your comment. The abstract has been completely rewritten based on your considerations.
Point 4: The Introduction sections addressed none updated literature review. I expected an introduction addressing the scientific mainstream of natural protected areas worldwide, but in reality, it is just a perspective of the authors based on very old literature.
Response 4: We thank you again for the comment. As mentioned above, this is a question derived from not having clearly expressed that it was an essay, which undoubtedly affects the expectations in its reading. We believe that the mentioned new updated references to support our perspective solve this problem in the context of an essay.
Point 5: The section Protected areas and cultural construction of Nature is a mere perspective, without a further approach of scientific and updated evidences of the usefulness of the protected areas worldwide.
Response 5: Thanks. Once again, we believe that it should be interpreted as an essay from a specific perspective whose aim is not to establish conclusive assertions, but to encourage reflection. In any case, we have expanded and updated the references and evidence on which these reflections are based (especially in the lines: 209-215; 221-225; 245-249; 250;253; 274-284; 300-322).
Point 6: If it is an article reflection about the protected areas, I miss quotations by Leopold and Rachel Carson, for instance, who were promoters of natural protected areas in the US.
Response 6: Thanks to this appreciation we have incorporated historical references that have improved the text (lines: 85-96; 153-171).
Point 7: The assertion in section 3, A protected area is considered the most successful instrument for nature conservation and the quotation of the Convention of Biological Diversity is disturbing. Actually, a protected area is not the most successful instrument but one of many other management instruments. Also, the authors fail in quoting the works by Callum Roberts, Tundy Agardy, and many others.
Response 7: Thank you for pointing out this error. We agree with the idea that the protected areas model is only an alternative for territorial management. We believe that it was an error in the way of expressing it. We really wanted to emphasize that on many occasions they are presented as the most successful instrument for nature conservation. We have modified the phrase and we believe that it is now better understood.
On the other hand, thanks to your considerations, as already mentioned above, we have expanded and updated the references.
Point 8: In sum, the work is a mixture of authors points of view and not a throughout review of updated and scientific evidences either in support or against what a natural protected area, either in terrestrial or marine environments.
Response 8: Thanks. We believe that not having sufficiently explained that it is an essay and the need to expand and update the bibliographical references are the two fundamental weaknesses of the work. We believe that we have significantly improved these issues.
We have followed your suggestions and incorporated them into the revised manuscript. We think the new version is now clearer.
Reviewer 2 Report
I will suggest to the authors to include a systematic literature review to improve the manuscript as for now if we read it we feel like it is missing some important ideas at the start. Like if authors are concerned with human activities in the protective areas, they should provide some more literature. I am still confused about what type of consideration is needed for the contribution of protected areas to sustainable development. This could be submitted as an essay, not a research article. Maybe the inclusion of some sustainable parameters and then the contribution of the protected areas to them will increase the readability of the paper.
Author Response
Thank you for your review.
We believe that one of the main problems with the manuscript has been that it was not sufficiently clear that it is an essay, not an experimental analysis paper. This error has undoubtedly influenced the expectations of its reading.
We think that in the new version It is clarified that it is an essay, and the aim of the paper is now clearer. Clarifications have been included in the abstract, introduction and final considerations (lines 8-13; 59-71; 250-253; 453-455). Furthermore, we have decided to replace the term 'contributions' with 'implications' in the title, since we believe that the former could cause confusion.
Based on their recommendations we have rewritten the manuscript and enhancing bibliographical references, updating our arguments based on more updated scientific literature throughout the manuscript.
Reviewer 3 Report
1) The information on methods applied is missing both in the Abstract and in the main text (on they are presented not so clear).
2) In my opinion, the author should in the conclusions emphasize the fact that since protected areas are in a way are a product of culture, then through the further development of culture, it is possible to further preserve the territory, ensuring a sustainable development.
Author Response
Point 1: The information on methods applied is missing both in the Abstract and in the main text (on they are presented not so clear).
Response 1: Thank you for your review.
We believe that one of the main problems with the manuscript has been that it was not sufficiently clear that it is an essay, not an experimental analysis paper. This error has undoubtedly influenced the expectations of its reading. We think that in the new version It is clarified that it is an essay, and the aim of the paper is now clearer. Clarifications have been included in the abstract, introduction and final considerations (lines 8-13; 59-71; 250-253; 453-455). Furthermore, we have decided to replace the term 'contributions' with 'implications' in the title, since we believe that the former could cause confusion.
Point 2: In my opinion, the author should in the conclusions emphasize the fact that since protected areas are in a way are a product of culture, then through the further development of culture, it is possible to further preserve the territory, ensuring a sustainable development.
Response 2: Thanks to your proposal we have reinforced throughout the manuscript the need to rethink the ontological perspectives of different cultures and how this aspect can affect the effectiveness of protected areas (lines 44-53; 59-71; 250-253; 311-322; 435-447). This was really the perspective on the implications of culture that we were trying to emphasize.
Reviewer 4 Report
I really liked this paper. Learned a lot and others will too. I must admit that I was slowed down every time I came across "Nature" (in quotes) and nature without quotes. I'm not sure I (always? Ever?) understood the distinction correctly (Is Mars a natural (no quotes) environment? Is "nature" only the human benefits? etc.). If there's some way the authors could help remove any vagueness that would probably help the flow of the manuscript. Also, I found the "...on the other hand..."s somewhat annoying. It's great to present both sides, but I was looking more for the benefit of distinctions made by the expert authors who have thought very deeply about the topic. Maybe the authors might consider these comments, but I would not insist on changes.
Author Response
Thank you very much for your comments. We are glad that you enjoyed our work.
Thanks for pointing out that the use of quotes and italics in Nature was confusing, we think we've improved this. Likewise, we have improved the flow of the manuscript based on your recommendations. We have rewritten the manuscript enhancing updating our arguments based on more updated scientific literature throughout the manuscript, and we think the new version is now clearer.